# Feature Fusion and Detection in Alzheimer’s Disease Using a Novel Genetic Multi-Kernel SVM Based on MRI Imaging and Gene Data

**DOI:** 10.3390/genes13050837

**Published:** 2022-05-07

**Authors:** Xianglian Meng, Qingpeng Wei, Li Meng, Junlong Liu, Yue Wu, Wenjie Liu

**Affiliations:** 1School of Computer Information and Engineering, Changzhou Institute of Technology, Changzhou 213032, China; mengxl@cit.edu.cn (X.M.); 18030722@czust.edu.cn (Q.W.); 19030410@czust.edu.cn (J.L.); 20030630@czust.edu.cn (Y.W.); 2School of Physics, Engineering and Computer Science, University of Hertfordshire, Hatfield AL10 9AB, UK

**Keywords:** Alzheimer’s disease, MRI imaging, gene, eigenvalue, genetic multi-kernel SVM, significant feature

## Abstract

Voxel-based morphometry provides an opportunity to study Alzheimer’s disease (AD) at a subtle level. Therefore, identifying the important brain voxels that can classify AD, early mild cognitive impairment (EMCI) and healthy control (HC) and studying the role of these voxels in AD will be crucial to improve our understanding of the neurobiological mechanism of AD. Combining magnetic resonance imaging (MRI) imaging and gene information, we proposed a novel feature construction method and a novel genetic multi-kernel support vector machine (SVM) method to mine important features for AD detection. Specifically, to amplify the differences among AD, EMCI and HC groups, we used the eigenvalues of the top 24 Single Nucleotide Polymorphisms (SNPs) in a *p*-value matrix of 24 genes associated with AD for feature construction. Furthermore, a genetic multi-kernel SVM was established with the resulting features. The genetic algorithm was used to detect the optimal weights of 3 kernels and the multi-kernel SVM was used after training to explore the significant features. By analyzing the significance of the features, we identified some brain regions affected by AD, such as the right superior frontal gyrus, right inferior temporal gyrus and right superior temporal gyrus. The findings proved the good performance and generalization of the proposed model. Particularly, significant susceptibility genes associated with AD were identified, such as *CSMD1*, *RBFOX1*, *PTPRD*, *CDH13* and *WWOX*. Some significant pathways were further explored, such as the calcium signaling pathway (corrected *p*-value = 1.35 × 10^−6^) and cell adhesion molecules (corrected *p*-value = 5.44 × 10^−4^). The findings offer new candidate abnormal brain features and demonstrate the contribution of these features to AD.

## 1. Introduction

Alzheimer’s disease (AD) and other forms of dementia severely affect a variety of cognitive functions, including memory. With the development of sequencing technology, scientists have conducted in-depth analyses of the association of genome-wide data and multimodal imaging phenotype data with the disease. This made it possible for scientists to characterize the abnormal genes and brain regions of AD. Genome-wide association analysis (GWAS), as an effective method to study the association between genetic data and phenotypic data, has been adopted to mine the genes associated with phenotypes through statistical methods [1]. Other scientists have developed new research methods for association analysis [2,3,4,5]. Some important single nucleotide polymorphisms (SNPs) might be missed only using GWAS, and the use of classical machine learning methods combined with GWAS and MRI images to screen genetic variants has become another research focus.

Rajeesh et al. [6] extracted 69 texture features from hippocampal MRI images of 133 AD patients and 146 healthy controls (HC), and used a support vector machine (SVM) as a benchmark to classify AD and normal people. They obtained a 93.6% classification accuracy and proved that the purpose of screening MRI images could be achieved by extracting the hippocampal texture characteristics of AD and normal people. Guenther et al. [7] constructed the maximum likelihood method in genetic association analysis and proved that this method could prevent bias and spurious signals in simulation studies and could unearth real association signals from spurious signals. Seo et al. [8] used methods such as machine learning, GWAS, linkage disequilibrium and principal component analysis to label the SNPs of specific flocks and constructed a combination of SNP markers. Then they used AdaBoost, random forests and decision trees for classification and achieved good classification results. Li et al. [9] constructed a neural network model using MRI images and used transfer learning to train the constructed model, demonstrating the relationship between non-invasive MRI and the development of AD for the first time. Huang et al. [10] applied a multi-kernel SVM to mine the white matter structural network features from mild cognitive impairment (MCI) and HC. Fidel et al. [11], Kinreich et al. [12] and Brabec et al. [13] applied machine learning methods to discover the features of AD and other diseases. In addition, Matthews et al. [14] discussed the current state of functional connectomics and envisaged greater potential to mine potentially significant information by combining different methods. However, there are still some limitations when mining GWAS results or MRI images using machine learning. For example, although genetic markers could be mined using GWAS results and machine learning methods, the same approach was not useful for mining abnormal brain regions. Using MRI images with machine learning methods can obtain excellent features and good classification accuracy for AD-HC. However, they did not perform well for EMCI-HC. Due to the huge amount of GWAS results, how to extract useful information from it, integrate it with MRI image information and apply it to further improve the accuracy of the machine learning methods remains one of the key challenges in the diagnosis of AD.

To bridge this gap, we proposed a novel feature construction method and a genetic multi-kernel SVM method to extract the important features that performed well in classification. Specifically, we used the *p*-value of the SNPs associated with AD to form a matrix and calculated the eigenvalues of this matrix. We then applied the eigenvalues and the original dataset to construct a new dataset. Subsequently, we proposed a genetic multi-kernel SVM model to extract important features from the resulting dataset. Finally, we used the extracted features to identify significant genes and analyzed the biological significance of these genes. We used the mild cognitive impairment (EMCI) dataset to verify the universality of our methods. The proposed feature construction method and genetic multi-kernel SVM model are shown in Figure 1.

## 2. Materials and Methods

### 2.1. Imaging and Gene Data

In this study, we applied MRI Imaging and gene data to conduct the experiment. Data from 922 subjects (HC:353, EMCI:273, AD:296) were downloaded from the Alzheimer’s Disease Neuroimaging Initiative (ADNI) (https://adni.loni.usc.edu/ (accessed on 15 May 2020). ADNI was launched in 2003 and provided the MRI, positron emission tomography (PET) and genetic data for HC, EMCI and AD. ADNI 1 was the first stage, and ADNI GO/2 was the second stage of ADNI. Data used in this study were downloaded from ADNI1/GO Month 6, ADNI 1/GO Month 12, ADNIGO Month 3 MRI and ADNI2 Year 1 visit studies. Table 1 presents the details of these subjects.

We preprocessed and segmented the MRI scans using voxel-based morphometry (VBM). The resulting images were normalized to the Montreal Neurological Institute (MNI) space. We then used an 8 mm full width at half maxima (FWHM) kernel to smooth the gray matter density (GMD) maps. The obtained maps were down-sampled to 61 × 73 × 61 to reduce the amount of data. Then, the Anatomical Automatic Labeling (AAL) atlas [15] was applied to obtain the coordinates of the brain regions.

The SNPs were selected using the method described in [16,17]. Briefly, based on the manufacturer’s protocol, Illumina GWAS arrays (610-Quad v1.0, OmniExpress-24 Kit or HumanOmni2.5-4v1) (Illumina, Inc., San Diego, CA, USA) and blood genomic DNA samples were applied for genotyping in downloaded subjects [18]. Using PLINK v1.9 [19], we extracted SNPs satisfying the following conditions: (1) on chromosome 1–22; (2) call rate ≥ 95%; (3) minor allele frequency ≥ 5%; (4) Hardy–Weinberg equilibrium test *p ≥* 1.0 × 10^−6^; (5) call rate of each participant ≥ 95%. A total of 5,574,300 SNPs passed the quality control.

Using the obtained results, we performed a GWAS (linear regression) in PLINK for each group. Age, gender, education and the top 10 principal components from population stratification analysis were included as covariates. We then performed Bonferroni correction on the results for multiple testing. 

### 2.2. Feature Construction

To extract the candidate features, we performed a weighted average on the three groups of images. Let $V_{mn}^{\prime }$ represent the vector of the AD-HC group. Then, the results were saved as matrices ***M***, ***N*** and ***O*** (***M*** represents AD, ***N*** represents HC and ***O*** represents EMCI). We defined the vector of one voxel as $\left(vm_{i}^{\prime },vn_{i}^{\prime }\right),\;where\;vm_{i}^{\prime }\in M,vn_{i}^{\prime }\in N$ and $V_{mn}=\left\{\left(vm_{1}^{\prime },vn_{1}^{\prime }\right),\left(vm_{2}^{\prime },vn_{2}^{\prime }\right),\ldots ,\left(vm_{k}^{\prime },vn_{k}^{\prime }\right)\right\}$ (k = 61 × 73 × 61 = 271,633) represented the vector of all voxels. After removing the value zero, we obtained a new vector $V_{mn}^{\prime }$ composed of 64,411 features. Similarly, $V_{mo}^{\prime }$ was the vector of the AD-EMCI group, $V_{no}^{\prime }$ was the vector of the EMCI-HC group, and the number of features in these two groups was 64,411.

For our binary classification, $V_{mn}^{\prime }$, $V_{mo}^{\prime }$ and $V_{no}^{\prime }$ were still not optimal. Therefore, we defined the upper and lower bounds of the number of features to filter features. The lower bound was obtained as follows: Equation (1) calculates the similarity between voxels in the 3 groups.
(1)$$\rho_{1}=\sqrt{(vm_{i}^{\prime }-vm_{j}^{\prime }{)}^{2}+{\left(vn_{i}^{\prime }-vn_{j}^{\prime }\right)}^{2}},\;\;\;\;\;\;\left(vm_{i}^{\prime },vn_{i}^{\prime }\right),\left(vm_{j}^{\prime },vn_{j}^{\prime }\right)\in V_{mn}^{\prime }\;\rho_{2}=\sqrt{(vm_{i}^{\prime }-vm_{j}^{\prime }{)}^{2}+{\left(vo_{i}^{\prime }-vo_{j}^{\prime }\right)}^{2}},\;\;\;\;\;\;\left(vm_{i}^{\prime },vo_{i}^{\prime }\right),\left(vm_{j}^{\prime },vo_{j}^{\prime }\right)\in V_{mo}^{\prime }\;\rho_{3}=\sqrt{(vn_{i}^{\prime }-vn_{j}^{\prime }{)}^{2}+{\left(vo_{i}^{\prime }-vo_{j}^{\prime }\right)}^{2}},\;\;\;\;\;\;\;\left(vn_{i}^{\prime },vo_{i}^{\prime }\right),\left(vn_{j}^{\prime },vo_{j}^{\prime }\right)\in V_{no}^{\prime }$$
where $vm_{i}^{\prime }$ and $vm_{j}^{\prime }$ (i,j=1,2,…,64,411) are the values of voxels in AD, $vn_{i}^{\prime }$ and $vn_{j}^{\prime }$ in HC, $vo_{i}^{\prime }$ and $vo_{j}^{\prime }$ in EMCI. $\rho_{1}$, $\rho_{2}$ and $\rho_{3}$ are the similarity between each pair. 

Let $C_{min}$ represent the lower bound with a value equal to the number of minimal $\rho_{i}$ and $C_{min}=132$. The upper bound was defined as $C_{max}=C_{min}+\sqrt{64,411}$ and $C_{max}\approx 386$. Thus, the number of features was 132,386.

For each group, we used the 24 genes [20] associated with AD to generate genetic features. First, we selected the top 24 SNPs of each gene to compute a matrix $M_{gene}$. Then, we applied the corresponding *p*-values of SNPs to construct the matrix $M_{p}$. Finally, we introduced Equation (2) to calculate the max feature $\lambda_{max}$.
(2)$$\left(\lambda E-M_{p}\right)x=0$$
where $\lambda_{max}=\max\left(\lambda \right)$, *E* is the unity matrix, and *x* is the eigenvector. $M_{p}$ is the *p*-value matrix of SNPs in AD, EMCI and HC groups.

We obtained 3 $\lambda_{max}$ and defined them as $\lambda_{AD}$, $\lambda_{EMCI}$ and $\lambda_{HC}$. Then, we applied the $\lambda_{AD}$, $\lambda_{EMCI}$ and $\lambda_{HC}$ to the 64,411 voxels to calculate fusion features. Let $M_{AD}$, $M_{EMCI}$ and $M_{HC}$ represent the 64,411 voxels of the AD, EMCI and HC groups. $S_{1},\;S_{2}$ and $S_{3}$, composed of fusion features, were defined as Equation (3).
(3)$$S_{1}=\left[\left[\lambda_{AD}\times M_{AD}\right],\;\left[\lambda_{HC}\times M_{HC}\right]\right]\;S_{2}=\left[\left[\lambda_{AD}\times M_{AD}\right],\;\left[\lambda_{EMCI}\times M_{EMCI}\right]\right]\;S_{3}=\left[\left[\lambda_{EMCI}\times M_{EMCI}\right],\left[\lambda_{HC}\times M_{HC}\right]\right]$$
where $S_{1}$, $S_{2}$ and $S_{3}$ represent fusion features matrix of the AD-HC group, AD-EMCI group and EMCI-HC, respectively. They are 649 × 64,411, 626 × 64,411 and 569 × 64,411, respectively.

### 2.3. Genetic Multi-Kernel SVM Construction

Using the HC and AD groups as examples, the multi-kernel SVM [21] method was applied to classify AD subjects from controls using ***S***_1_. Our multi-kernel SVM was based on traditional SVM, which used a linear combination of multiple kernel functions to fuse and then trained a SVM classifier based on the fused kernel.

The fused kernel function can be written as a linear combination of basic kernels [22], as defined in Equation (4).
(4)$$K\left(x_{i},x_{j}\right)=w_{1}K_{1}\left(x_{i},x_{j}\right)+w_{2}K_{2}\left(x_{i},x_{j}\right)+w_{3}K_{3}\left(x_{i},x_{j}\right)$$
where w is the weight on the corresponding basic kernel $K\left(x_{i},x_{j}\right)$. $K_{1}$, $K_{2}$ and $K_{3}$ are different kernel functions.

We selected basic kernels, such as linear, polynomial and radial basis function (RBF), to compose the final kernel function. Then, the optimal w of each kernel was determined using genetic evolution. Specifically, the initial population consisting by $\left(w_{1},w_{2},w_{3}\right)$ was generated randomly, and the parents were randomly selected from the initial population. Then, the offspring were obtained by the crossover of the parents, and the introduction of new data was realized by setting a variogram. The decision function [23] in the classification is written as Equation (5):(5)$$f\left(x\right)=sgn\left\{\sum_{i=1}^{n}\alpha_{i}^{*}y_{i}\sum_{d}K\left(x_{i},x_{j}\right)+b^{*}\right\}$$
where $\alpha_{i}$ is the Lagrange multiplier, * is the dot product of the vector. sgn represents the symbolic function corresponding to the classification label. $y_{i}$ is the prediction result, and *b* is the intercept in the linear equation. The decision function f(x) has only two output values: −1 and +1.

The classification accuracy of AD and HC was defined as Equation (6).
(6)$$Acc=N_{T}/N$$
where *Acc* is the classification accuracy, $N_{T}$ is the number of correct classifications, and N is the total number of subjects in ***S****_1_*.

The genetic multi-kernel SVM was constructed by repeating the above process.

***S***_1_ was randomly split as $S_{train}:S_{valid}:S_{test}=6:2:2$. To obtain the optimal $\left(w_{1},w_{2},w_{3}\right)$, we first adjusted the parameters of the genetic process to determine the optimal population size and genetic evolution iterations using $S_{train}$ and $S_{valid}$. By setting the optimal population size and genetic evolution iterations to (200, 1000) and (20, 100) and the step size to 200 and 20, all parameter combinations were traversed to find the optimal one.

We then used the optimal parameter combination with $S_{train}$ and $S_{test}$ to find the best $\left(w_{1},w_{2},w_{3}\right)$ and features with high classification accuracy. To ensure the reproducibility of the experiments, we performed 10 independent experiments following this process.

We also applied the single kernel SVM (linear, polynomial or RBF) and the original dataset (not adjusted using genetic data) to compare with the proposed method in this paper to verify the superiority of our method.

### 2.4. Gene Identification and Biological Significance Assessment

Through the above steps, we obtained three sets of features from the AD-HC group, AD-EMCI group and EMCI-HC group. We calculated the brain regions where the features were located using their coordinates and the AAL atlas. Then, using the features from each group as a phenotype, we performed a GWAS (linear regression) in PLINK with the covariates described in Section 2.1. Then, we applied the effective chi-squared test (ECS) method [24] to calculate *p*-values for genes and Bonferroni correction for multiple testing. The resulting genes with a corrected *p*-value < 0.001 were selected to identify the significant genes in all 3 groups and the significant specific genes in each group. Furthermore, they were also used to assess biological significance through pathway analysis [25].

## 3. Results

### 3.1. Results of Parameter Optimization

Initially, *n* features (132≤n≤386) were randomly extracted from the candidate features. With $w_{1}=w_{2}=w_{3}=1$, we applied these features and weights in a genetic multi-kernel SVM to adjust the optimal population size and genetic evolution iterations. First, we randomly selected features and $S_{train}$ and $S_{valid}$ randomly. Then, as described in Section 2.3, we traversed all parameter combinations. To determine the best parameter combination, we repeated the steps above 100,000 times. The peak classification accuracy of each parameter combination is shown in Figure 2.

As shown in Figure 2, the peak of the classification accuracy in the AD-HC group is at the node of 800–100, and the corresponding accuracy is 90.84%. Therefore, the best combination of population size and generation times for AD-HC was (800, 100). Similarly, the best combinations for AD-EMCI and EMCI-HC were (600, 100) and (800, 40), respectively. Their corresponding accuracy results were 90.18% and 80.6%.

### 3.2. Comparison with Other Methods

Using the parameter combinations obtained, we conducted 10 independent repeat experiments in each group to mine the optimal weights of the kernel functions. The results are shown in Figure 3.

From Figure 3a for AD-HC, we observed that all the results converge. The best accuracy was found in the 6th and 7th experiments, and the peak accuracy was 93.8%. From Figure 3b AD-EMCI, we observed that most of the experiments converged around generation 500, and the results of the 6th and 7th experiments did not converge. The best accuracy was found in the 9th and 10th and the peak accuracy was 94.73%. The best combination was 0.9897, 0.19483, 0.94445. From Figure 3c for EMCI-HC, we observed that the results of the 8th did not converge; and in this experiment, a suitable maximum value was not found by the genetic algorithm. In the other 9 experiments, suitable values were found, and the best results were obtained in the 3rd and 6th experiments. The accuracy of these two experiments was 85.6%, and their corresponding generation times were around 600. The best $\left(w_{1},w_{2},w_{3}\right)$ was (0.55105, 0.66182, 0.63488). This indicates that parents with high accuracy had a greater chance of passing on their good genes to their offspring. The peak accuracy of the EMCI-HC group was lower than that of the other two groups. This may be caused by the lower difference between the EMCI and HC. Finally, we applied the optimal parameters to the genetic multi-kernel SVM and obtained 145 features in AD-HC, 199 features in AD-EMCI and 315 features in EMCI-HC.

Using the best $\left(w_{1},w_{2},w_{3}\right)$ obtained, we conducted 10 independent repeat experiments with 5 methods, including linear kernel SVM, poly kernel SVM, radial basis function (RBF) kernel SVM, multi-kernel SVM and genetic multi-kernel SVM. The results are shown in Figure 4.

As shown in Figure 4, although the maximum and minimum accuracy of our model varied widely, peak accuracy was produced by our model for all three groups. The accuracy of RBF kernel SVM is the lowest among the 5 tested models. The application of the RBF kernel function may have reduced the performance of the genetic multi-kernel SVM, resulting in a large fluctuation in its accuracy. The high accuracy of genetic multi-kernel SVM came from a specific fusion of different kernel functions. The peak accuracy in AD-HC, AD-EMCI and EMCI-HC was 93.8%, 94.7% and 85.6%. 

### 3.3. Identification of Brain Regions and Genes 

Using the extracted features and their coordinates, we calculated the brain regions where they were located. The top 10 brain regions with the most features are listed in Table 2. The right superior frontal gyrus (Frontal_Sup_R), the right inferior temporal gyrus (Temporal_Inf_R) and the right superior temporal gyrus (Temporal_Sup_R) were the most frequently highlighted brain regions for the three groups, respectively.

We also performed GWAS in each group and calculated *p*-values for genes. Then, we used the top 10 Bonferroni-corrected genes for gene identification. The significant common genes in the three groups are listed in Table 3. Our study also demonstrated that pathogenic genes, such as *CSMD1*, *RBFOX1*, *PTPRD*, *CDH13* and *WWOX*, were significantly related to AD [26,27,28,29,30]. The significant specific genes in each group are listed in Table 4.

### 3.4. Biological Significance Assessment

The corrected genes with *p*-values < 0.001 were also applied for pathway analysis. The top 15 pathways and their corrected *p*-values for each group are shown in Figure 5 [41].

As shown in Figure 5, eight pathways were present in all three groups, and 5 pathways were present in both the AD-HC group and the EMCI-HC group. Only 1 pathway (Glutamatergic synapse) was in both the AD-EMCI group and the AD-HC group. We also found that 6 pathways from the AD-EMCI group did not intersect with the other two groups, while there was only 1 in the AD-HC group and 2 in the EMCI-HC group that did not intersect with the other groups. Moreover, the corrected *p*-values of pathways in the AD-EMCI group were superior to the other two groups.

In addition, we counted the distribution of pathways with a corrected *p*-value < 0.001 in the three groups. The results are shown in Figure 6 [41].

From Figure 6, we observed that most of the pathways were found in all three groups. Considering that MCI is a precursor state to AD, it is reasonable to find the same pathways in multiple groups. The numbers of pathways shared by only two groups are 0, 1 and 1. Interestingly, there were 3 pathways that are present only in AD-HC and 1 pathway only in EMCI-HC. However, there were 15 pathways present only in AD-EMCI, significantly exceeding the number of exclusive pathways in the other two groups. The number of top genes used for pathway analysis among the three groups were 1 (AD-EMCI and EMCI-HC), 9 (AD-HC and EMCI-HC) and 10 (AD-HC and AD-EMCI). The difference in the number of genes was small, but the number of identified pathways was far larger. The reason for this may lie in the different top genes. 

## 4. Discussion

In this paper, we propose a novel feature construction method and a novel feature detection model. In particular, we applied the eigenvalues of the SNP matrix to the original dataset and obtained promising classification results. 

As shown in Figure 2, the optimal generation times of the three data groups were either 600 or 800. This indicates that lower evolution times may not yield good classification results, while higher evolution times may lead to a drop in accuracy due to overfitting and random mutations. Figure 3 shows that the best accuracy of each experiment occurred around 600 evolution times. This is consistent with previous findings. In comparison experiments with the other methods, although good classification accuracy was achieved by the single kernel SVM, the multi-kernel SVM achieved better. This confirms that the multi-kernel SVM is able to take advantage of the differences among the multiple kernel functions and apply them in the model training. The addition of gene data improved the classification accuracy of the multi-kernel SVM. This proves that the genetic data contains information useful for classification. How to scientifically and effectively integrate genetic data is worth further investigation. In other papers, genetic data has been used to construct fusion features [42,43,44], leading to satisfactory accuracy. In this paper, we extracted 24 genes associated with AD [20] and calculated the eigenvalues of the matrix formed by the corresponding SNPs. The maximum eigenvalue was applied to construct the new dataset and higher accuracy was obtained using the resulting dataset. The eigenvalue amplifies the differences in the imaging data among AD, EMCI and HC groups. By only adjusting the parameters in our model, we obtained satisfactory accuracy for all three different groups. This proves that our proposed model has excellent generalizability.

For the extracted features, we calculated the number of times they each appeared in 90 brain regions. Then, we found that the most frequently highlighted brain regions were the right superior frontal gyrus (Frontal_Sup_R) in AD-HC, the right inferior temporal gyrus (Temporal_Inf_R) in AD-EMCI, and the right superior temporal gyrus (Temporal_Sup_R) in EMCI-HC. The regional homogeneity value increased in the superior frontal gyrus and affected memory and cognition [45]. The inferior temporal gyrus is involved in cognitive impairment and immediate visual memory by forming the inferior longitudinal fasciculus [46]. Although the right superior temporal gyrus was not found to be a significant region for AD, the fractional amplitude of low-frequency fluctuation value was increased in the left superior temporal gyrus and by jointly combining the regional homogeneity value, the researchers could explore the mechanism of the brain [14,45].

We found five AD-related genes in the three groups (AD-EMCI, AD-HC, and EMCI-HC; see Table 3). *CSMD1* (*p*-value = 1.02583 × 10^−29^) has been associated with AβPP metabolism and affected AD pathogenesis [26]. *RBFOX1* (*p*-value = 5.84303 × 10^−22^) is expressed in the brain and related to the brain function that had been confirmed to be related to AD and MCI [27]. It can be seen from Table 4 that the risk AD genes, such as *MEIS2* (*p*-value = 3.0419 × 10^−147^), *DLGAP2* (*p*-value = 3.57803 × 10^−19^) and *MAGI2* (*p*-value = 8.108022 × 10^−15^) have been identified in the AD-HC group [31,32,33]. We suggest that *MIR8063* (*p*-value = 1.827314 × 10^−221^) may be associated with susceptibility to AD. Furthermore, some genes associated with AD, including *PRKN*, *LRP1B*, *ASIC2*, *PRKG1*, *PTPRT*, *NELL1* and *AGBL1,* were also successfully identified.

By analyzing Figure 5 and Figure 6, we found that when the threshold was set at 0.001, the pathways in the EMCI-HC group and AD-HC group were fewer than in the AD-EMCI group. Except for the pathways in all three groups, there was no intersection between the AD-HC group and the EMCI-HC group. This suggests that the shared pathways were prominent pathways for AD, while the distinct pathways were specific pathways of distinct groups that correlated with the extent of AD pathology. In addition, the AD-EMCI group had the most significant pathways and the most gene categories, indicating that the AD-EMCI group also had the most SNP categories. We speculated that the AD-EMCI group-specific genes and pathways might be in the transition state from HC to AD, and that their changes led to the transition from EMCI to AD.

For the pathways common in the three groups, the calcium signaling pathway (hsa04020, corrected *p*-value = 1.35 × 10^−6^) is a significant pathway. Ca^2+^ signaling has a certain regulatory effect on signal propagation in vivo. By regulating Ca^2+^, Calcium signaling pathway plays a role in the synaptic function of Aβ and is associated with early AD [47]. For example, high Ca^2+^ was found in neurites surrounding β-amyloid [48], and the application of Aβ activated the N-methyl-D-aspartate receptors and led to the Ca^2+^ rise [49]. Elevated *RYR2* (corrected *p*-value = 6.77 × 10^−7^ in AD-EMCI, corrected *p*-value = 1.47 × 10^−5^ in AD-HC and corrected *p*-value = 9.76 × 10^−3^ in EMCI-HC) and *RYR3* (corrected *p*-value = 6.2 × 10^−6^ in AD-EMCI, corrected *p*-value = 2.22 × 10^−7^ in AD-HC and corrected *p*-value = 4.4 × 10^−6^ in EMCI-HC) expression levels enhanced the Ca^2+^ release and caused the Ca^2+^ signal dysregulation in AD [50,51,52].

For the cell adhesion molecules (CAMs, corrected *p*-value = 5.44 × 10^−4^) pathway in the EMCI-HC group, increased NCAM expression was found in the hippocampus, decreased expression in the frontal/temporal cortex, and increased CSF levels [53,54,55,56]. Integrin unit (IU) β1 was involved in fibrillar Aβ-mediated microglia internalization [57], IUα4 was found near Aβ plaques [58], and IUβ3 was adjacent to Aβ plaques [59]. For the 3 pathways that presented only in the AD-HC group (Renin secretion, corrected *p*-value = 2.4 × 10^−4^; Apelin signaling pathway, corrected *p*-value = 3.29 × 10^−4^; Adrenergic signaling in cardiomyocytes, corrected *p*-value = 4.04 × 10^−4^) only in the AD-HC group, hypertension was associated with AD, and renin secretion was a pathway that regulated cerebral blood flow [60]. Among AD patients, the level of apelin in the blood significantly decreased [61], which has a positive effect on cognitive memory [62,63]. The adrenergic signaling in cardiomyocytes regulated Ca^2+^ in vivo by acting on cardiac muscle contraction and further on the calcium signaling pathway (https://www.genome.jp/kegg-bin/show_pathway?hsa04261). Changes in these pathways might lead to the transition from EMCI to AD.

## 5. Conclusions

In this study, we fused the voxel-based features extracted from MRI imaging data with the eigenvalue of SNPs and proposed a novel genetic multi-kernel SVM for AD detection. Our model was superior to other methods, including traditional single kernel SVM and standard multi-kernel SVM. We evaluated the generalizability of our model in different data groups, which can also be evaluated in other diseases in the future. Moreover, we analyzed the extracted features and showed that our method provides a new strategy for imaging genetics analysis. Some AD-related brain regions, such as the right superior frontal gyrus, right inferior temporal gyrus and right superior temporal gyrus, were found. We have identified more robust and stable AD-related genes, including *CSMD1*, *RBFOX1*, *PTPRD*, *CDH13* and *WWOX*. Our investigation shows that the calcium signaling pathway, cell adhesion molecule pathway, and oxytocin signaling pathway affected the development of AD. All of these findings contribute to a better understanding of the pathological changes in the course of AD.

## Figures and Tables

**Figure 1 genes-13-00837-f001:**
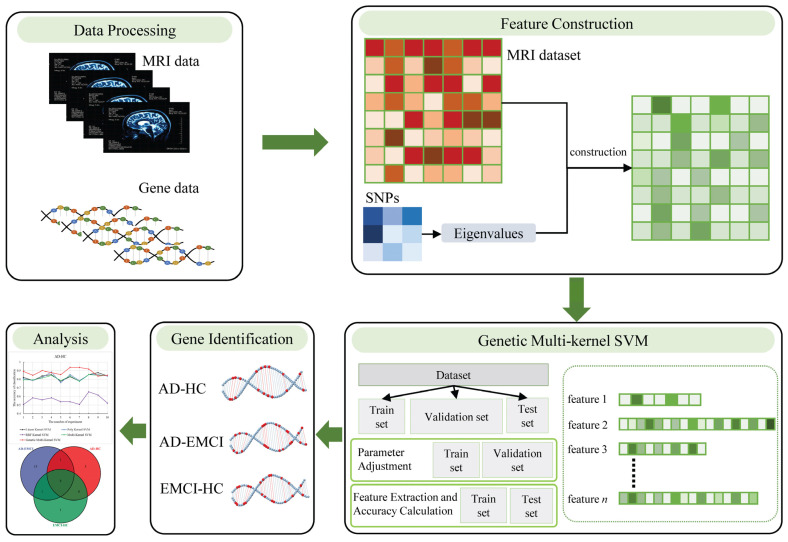
The proposed feature construction method and genetic multi-kernel SVM model.

**Figure 2 genes-13-00837-f002:**
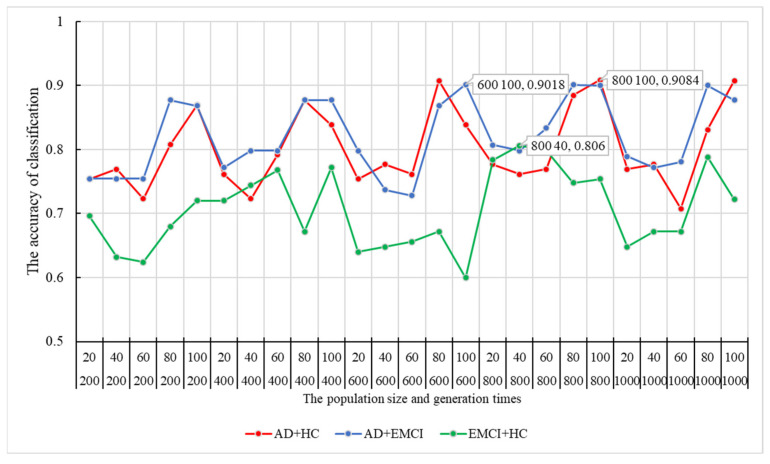
The optimal population size and generation times and their corresponding classification accuracy. HC = healthy control; EMCI = Early Mild Cognitive Impairment; AD = Alzheimer’s disease.

**Figure 3 genes-13-00837-f003:**
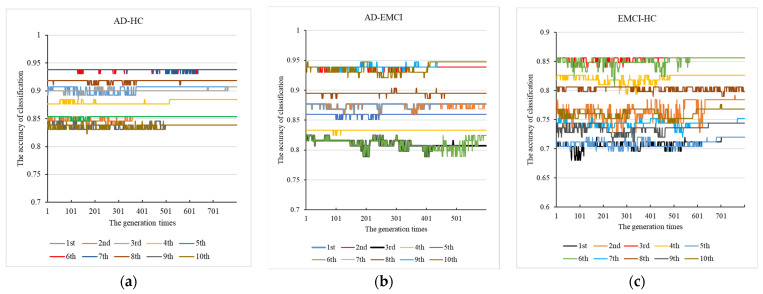
The 10 independent repeat experiments with (**a**) AD-HC, (**b**) AD-EMCI and (**c**) EMCI-HC.

**Figure 4 genes-13-00837-f004:**
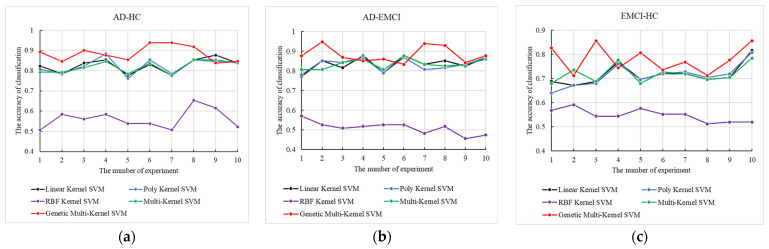
The 10 independent repeat experiments of the 5 methods on (**a**) AD-HC, (**b**) AD-EMCI and (**c**) EMCI-HC.

**Figure 5 genes-13-00837-f005:**
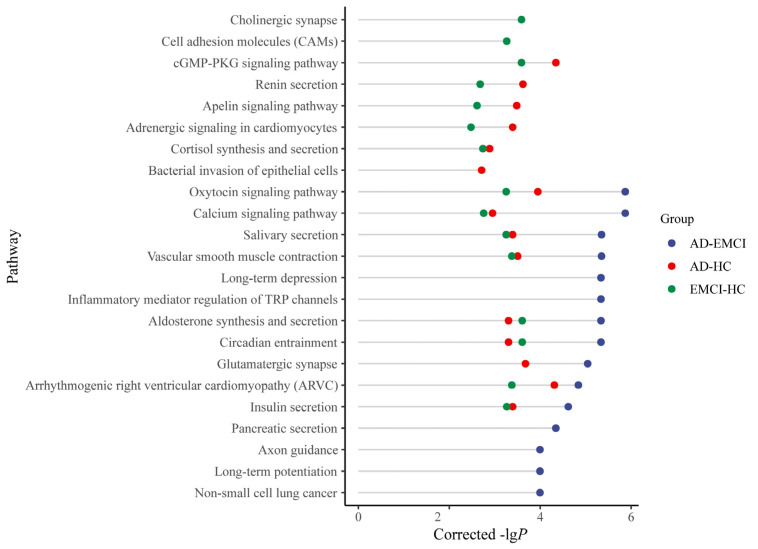
The top 15 pathways of the AD-HC group, AD-EMCI group and EMCI-HC group.

**Figure 6 genes-13-00837-f006:**
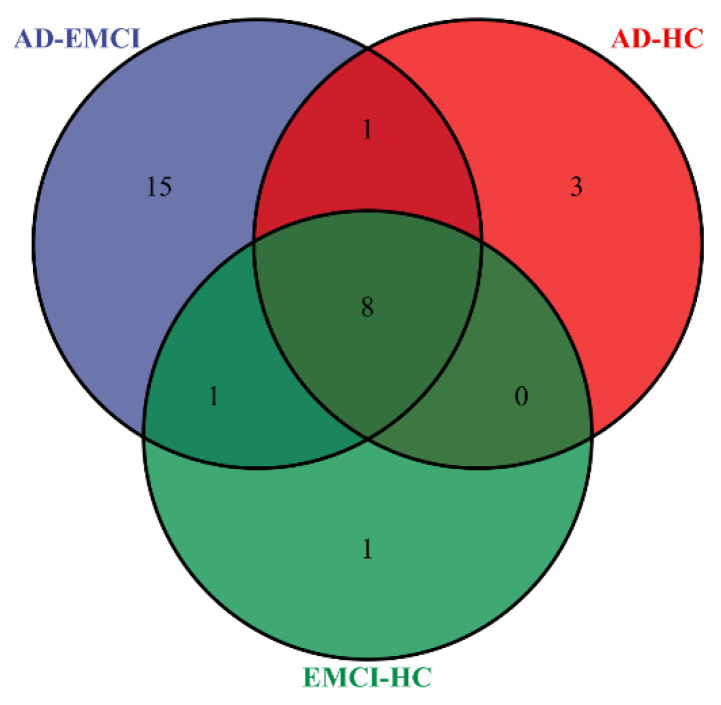
The distribution of pathways with the corrected *p*-value < 0.001 in the AD-HC group, AD-EMCI group and EMCI-HC group.

**Table 1 genes-13-00837-t001:** Subject characteristics. HC = healthy control; EMCI = Early Mild Cognitive Impairment; AD = Alzheimer’s disease; M/F = male/female; Edu = education; sd = standard deviation.

Subjects	HC	EMCI	AD	*p*
Number	353	273	296	-
Gender (M/F)	187/166	153/120	166/130	<0.001
Age (mean ± sd)	72.2 ± 7.6	71.3 ± 7.1	75.1 ± 5.5	<0.001
Edu (mean ± sd)	16.1 ± 2.7	16.1 ± 2.6	16.3 ± 2.6	<0.001

**Table 2 genes-13-00837-t002:** The top 10 brain regions with the most features.

AD-HC		AD-EMCI		EMCI-HC	
Brain Region	Number of Features	Brain Region	Number of Features	Brain Region	Number of Features
Frontal_Sup_R	9	Temporal_Inf_R	7	Temporal_Sup_R	6
Frontal_Mid_L	5	Precuneus_R	6	Frontal_Sup_L	5
Lingual_R	5	Frontal_Mid_L	5	Frontal_Inf_Orb_L	5
SupraMarginal_R	5	Precuneus_L	5	Frontal_Sup_Medial_L	5
Temporal_Mid_L	5	Postcentral_L	4	Calcarine_R	5
Frontal_Sup_L	4	Temporal_Sup_R	4	Fusiform_L	5
Frontal_Mid_R	4	Frontal_Mid_R	3	SupraMarginal_L	5
Lingual_L	4	Calcarine_L	3	Precuneus_R	5
Fusiform_L	4	Occipital_Mid_L	3	Temporal_Mid_L	5
Postcentral_R	4	Occipital_Mid_R	3	Temporal_Inf_R	5

**Table 3 genes-13-00837-t003:** Significant genes in the three groups.

Genes	AD-HC	AD-EMCI	EMCI-HC	References
	*p*-Value	*p*-Value	*p*-Value	
*CSMD1*	2.998108 × 10^−36^	1.02583 × 10^−29^	1.61113 × 10^−35^	Parcerisas et al. [26]
*RBFOX1*	5.84303 × 10^−22^	1.37062 × 10^−20^	6.3792 × 10^−26^	Raghavan et al. [27]
*PTPRD*	3.43579 × 10^−21^	3.81205 × 10^−24^	1.52404 × 10^−26^	Uhl et al. [28]
*CDH13*	5.58042 × 10^−20^	1.85248 × 10^−14^	6.10705 × 10^−13^	Liu et al. [29]
*WWOX*	7.1123 × 10^−17^	2.9447 × 10^−20^	2.46024 × 10^−22^	Hsu et al. [30]

**Table 4 genes-13-00837-t004:** Significant specific genes in each group.

Group	Gene	*p*-Value	References
AD-HC	*MIR8063*	1.827314 × 10^−221^	-
	*MEIS2*	3.0419 × 10^−147^	Huang et al. [31]
	*DLGAP2*	3.57803 × 10^−19^	Ouellette et al. [32]
	*MAGI2*	8.108022 × 10^−15^	Kim et al. [33]
AD-EMCI	*PRKN*	2.52709 × 10^−15^	Panda et al. [34]
	*LRP1B*	1.50983 × 10^−13^	Shang et al. [35]
	*ASIC2*	4.2832 × 10^−13^	Kreple et al. [36]
	*PRKG1*	5.64312 × 10^−13^	Koran et al. [37]
EMCI-HC	*PTPRT*	1.11037 × 10^−14^	Ben et al. [38]
	*NELL1*	1.14303 × 10^−12^	James et al. [39]
	*AGBL1*	2.88479 × 10^−11^	Dong et al. [40]

## Data Availability

Data used for this study were provided from ADNI studies via data sharing agreements that did not include permission to further share the data. Data from ADNI are available from the ADNI database (adni.loni.usc.edu) (accessed on 15 May 2020) upon registration and compliance with the data usage agreement.

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
