# Peer review of "Feature Fusion and Detection in Alzheimer’s Disease Using a Novel Genetic Multi-Kernel SVM Based on MRI Imaging and Gene Data"

_genes, 2022, doi:10.3390/genes13050837_

Round 1
Reviewer 1 Report
The paper describes feature fusion and detection in Alzheimer's disease using a novel genetic multi-kernel SVM based on MRI imaging and gene data. It is well written and contains all the relevant paper sections. they constructed neural networks and used complex mathematical equations and 6 figures and 1 table. The data is available at ADNI.
comment: the authors can add citation to the following paper:
Clinical Concepts Emerging from fMRI Functional Connectomics. Neuron. 2016 Aug 3;91(3):511-28. doi: 10.1016/j.neuron.2016.07.031.PMID: 27497220Author Response
Please see the attachment.
Reviewer 2 Report
General concept comments
The article discussed only genetic information that proved valuable for classification. Given that this method uses both MRI and genetic data, MRI information is neglected. Nothing was said about the MRI features that were extracted. comparison. Indeed to what extent are genetic features superior or do they complement MRI features?
Article is quite difficult to follow and read, mostly due to the language. A language review would be beneficial for the flow and understanding of the article.
Specific comments
Introduction mentions images (line 42, 46, 60, 62, 63).
Line 41: SNP abbreviation not explained
Line 60 to 62: Repeated sentence.
Line 72: abbreviation not explained (EMCI)
Line 80: You mention EMCI which are part of ADNI GO/2. Additional information on the exact database needs to be provided.
Line 120: So the number of features extracted from MRI images were between 132 and 386? The vectors from MRI features seems to be forgotten when combining them with SPN data.
Why are not the restricted features used? When S matrices are constructed all voxels are used.
Line 128: You only mention that you created fusion features for AD and HC group, yet later you provide combined matrix for EMCI also. I’m assuming that sentence in lines 128 and 129 is redundant?
Line 130: probably error; voxels of EMCI group
Line 140: Which S? You specify 3 datasets S (1, 2 and 3). What exactly does S dataset represents?
Line 148: RBF abbreviation not explained
Line 162: What does total number of dataset mean? Size of dataset?
Line 163: Isn’t the genetic data incorporated in the S matrices? Which matrices are used for multi-kernel SVM if genetic data is not considered?
Line 173: What exactly is the original dataset?
Line 180: abbreviation ECS not explained
Line 237: Not sure which number corresponds to which group
Round 2
Reviewer 2 Report
To my disapppointment, my earlier, even major comments were not or only scantily addressed. This is not acceptable. I add herewith a few new comments to be added to my previous ones.
Comments after first revision
- Abstract needs revision, because additional information was added about MRI findings.
- Introduction still contains vague mentions of images (e.g. line 42, 63). This needs to be specified as MRI images, otherwise the reader will be left to speculate about the modality of the images.
- The additional sentence in lines 61 and 62 does not add much to the text. What are the challenges that Matthews et al. mention? How does it overlap with limitations you are mentioning? Is the main limitation how to organically combine GWAS, machine learning and MRI? In that case it is technical, not conceptual.
- The additional text on the ADNI database is just a description of it. Instead, for the sake of reproducibility, it is important to note from which cohort/study your data came. You also mention DTI data but you did not use this modality, at least nothing is reported about it in the manuscript.
- Sentences in lines 133 to 136 should be written concisely. There is much redundancy (all groups used the same number of voxels).
- In the discussion, additional information was added on MRI findings. This needs to be represented in the Results section, otherwise it is not clear that this is the information you acquired from your analysis. In addition, in the text region labels (e.g. Frontal_Sup_R) are used which should be written as regions that readers will recognize and understand (e.g. frontal superior gyrus). It is also no clearly written what were your findings and that was done by other researchers. You have not used any cognitive measures, yet you state that there are observable effects (e.g. line 303-305).
Round 3
Reviewer 2 Report
The language has not improved, also my major points were not addressed.
1. Abstract needs revision, because additional information was added about MRI findings.
2. Introduction still contains vague mentions of images (e.g. line 42, 63). This needs to be specified to be MRI images, otherwise the reader will be left to speculate about the modality of the images.
3. The additional sentence in lines 61 and 62 does not add much to the text. What are the challenges that Matthews et al. mentions? How does it overlap with limitations you mention? Is the main limitation on how to organically combine GWAS, machine learning and MRI?
4. The additional text on ADNI database is just the description of ADNI. For the sake of reproducibility, it is important to note from which cohort/study your data came from. You also mentioned DTI data. Did you also use this modality?
5. Sentences in lines 133 to 136 should be written concisely. There is much redundancy (all groups used the same amount of voxels).
6. In discussion, additional information was added on MRI findings. This needs to be represented in Results section, otherwise it is not clear that this is the information you acquired from your analysis. In addition, in the text region labels (e.g. Frontal_Sup_R) are used which should be written as regions that readers will recognize and understand (e.g. frontal superior gyrus). It is also no clearly written what were your findings and that was done by other researchers. You have not used any cognitive measures, yet you state that there are observable effects (e.g. line 303-305).
